# Silver/chiral pyrrolidinopyridine relay catalytic cycloisomerization/(2 + 3) cycloadditions of enynamides to asymmetrically synthesize bispirocyclopentenes as PDE1B inhibitors

Jing Jiang[1], Jin Zhou[1], Yang Li[1], Cheng Peng[1], Gu He[2], Wei Huang[1], Gu Zhan [1✉] & Bo Han [1✉]

Significant progress has been made in asymmetric synthesis through the use of transition metal catalysts combined with Lewis bases. However, the use of a dual catalytic system involving 4-aminopyridine and transition metal has received little attention. Here we show a metal/Lewis base relay catalytic system featuring silver acetate and a modified chiral pyrrolidinopyridine (PPY). It was successfully applied in the cycloisomerization/(2 + 3) cycloaddition reaction of enynamides. Bispirocyclopentene pyrazolone products could be efficiently synthesized in a stereoselective and economical manner (up to >19:1 dr, 99.5:0.5 er). Transformations of the product could access stereodivergent diastereoisomers and densely functionalized polycyclic derivatives. Mechanistic studies illustrated the relay catalytic model and the origin of the uncommon chemoselectivity. In subsequent bioassays, the products containing a privileged drug-like scaffold exhibited isoform-selective phosphodiesterase 1 (PDE1) inhibitory activity in vitro. The optimal lead compound displayed a good therapeutic effect for ameliorating pulmonary fibrosis via inhibiting PDE1 in vivo.

[1] State Key Laboratory of Southwestern Chinese Medicine Resources, School of Pharmacy, Chengdu University of Traditional Chinese Medicine, Chengdu, Sichuan 611137, P.R. China. [2] State Key Laboratory of Biotherapy and Department of Pharmacy, West China Hospital Sichuan University, Chengdu 610041, P.R. China. ✉email: zhangu@cdutcm.edu.cn; hanbo@cdutcm.edu.cn

Developing highly selective, efficient, and economical synthetic strategies is an essential target in modern chemistry. Recently, both the multienzymatic systems and the multicatalysis of small molecules have received extensive attention due to their advantages over conventional methods using a single catalyst[1–18]. On the one hand, they can unify different bond activation modes by applying different types of catalysts. On the other hand, these processes are able to produce complex products from simple materials without the isolation of intermediates. Therefore, developing new multicatalytic systems is expected to achieve a more efficient and environmentally friendly organic synthesis, which features short time, low cost, and simple operation.

Significant advances have been made in this area by combining transition metal catalysts with organocatalysts[19–29]. Chemists have exploited cooperative and relay catalytic systems involving a transition metal with a Lewis base and demonstrated their great potential in asymmetric catalysis. In these processes, tertiary amines[30–33], phosphines[34,35], N-heterocyclic carbenes (NHCs)[36–39], and isothioureas[40–54] were successfully utilized as Lewis base catalysts (Fig. 1A). 4-aminopyridine such as 4-dimethylaminopyridine (DMAP), represents a special class of Lewis base catalysts[55–59]. They often exhibit excellent catalytic activity and unique selectivity due to their highly nucleophilic planar core structure. Besides, compared with other Lewis base catalysts, their aminopyridine skeleton is more modifiable, which is conducive to modular design and structural modifications[60–72]. However, in sharp contrast with other Lewis bases, the dual catalytic system of 4-aminopyridine with transition metal has been rarely studied[73,74], possibly because it can strongly coordinate to the metal center and deactivate the metal complex. Therefore, we envision developing a dual catalytic system combining transition metals with chiral 4-aminopyridines. It could be a powerful tool in asymmetric synthesis if the challenging issues of compatibility and stereocontrol were addressed.

The cycloisomerization/(3 + n) and (4 + n) cyclization cascade reactions of yne-enones and enynamides have emerged as important methods to construct diverse fused furans (Fig. 1B)[75–92]. Elegant catalytic systems and different cyclization reactions were successfully developed by the groups of Zhang, Ma, Chi, Zhou, Liu, Deng and others[93–100]. For instance, the Chi group reported the synthesis of furan-fused lactams via cycloisomerization/(4 + 2) cycloadditions by gold/NHC relay catalysis[93]. Zhao et al. achieved the gold/palladium relay catalytic cycloisomerization/(4 + 5) cycloaddition[94]. In these reports, the process tends to give the aromatic furan products through cascade (3 + n) or (4 + n) cyclization. However, the cycloisomerization/(2 + n) reaction has not been reported. This challenge might result from the steric hindrance of the cycloaddition step and the better thermodynamic stability of the aromatic products.

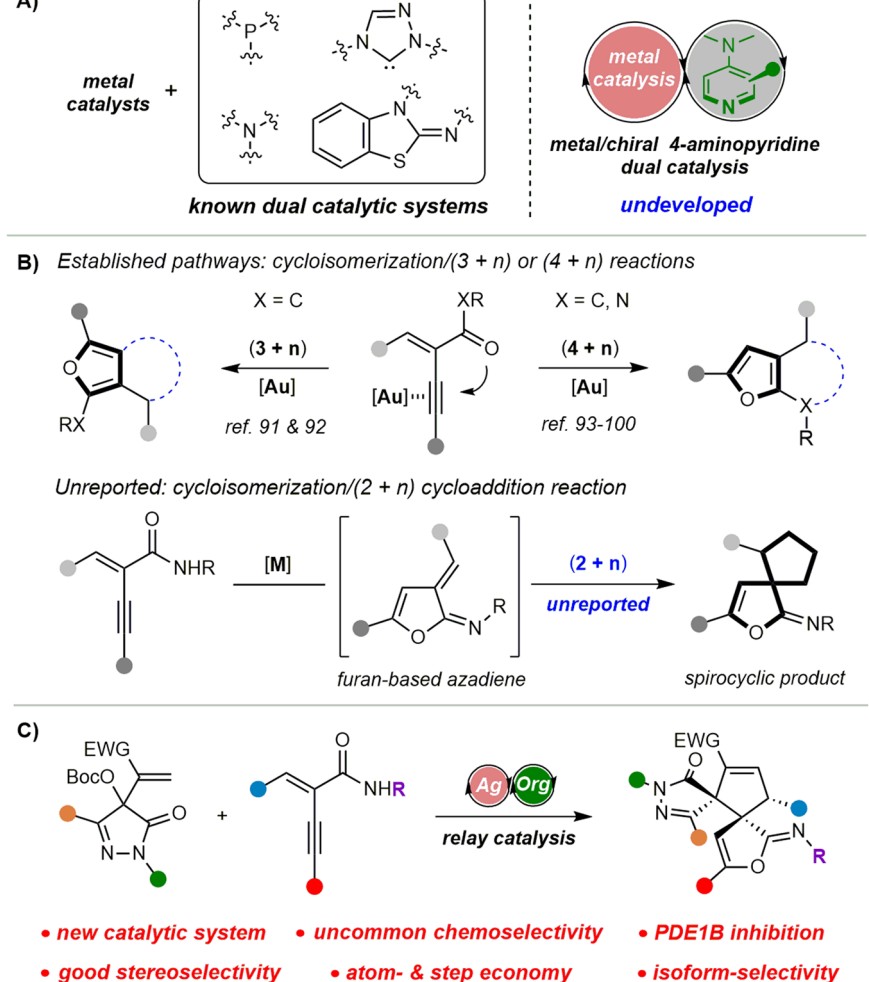

**Fig. 1 Metal/chiral 4-aminopyridine dual catalytic system and its application in cycloisomerization/(2 + 3) cycloaddition cascade reactions of enynamides. A** Known and undeveloped metal/organic Lewis base dual catalytic systems. **B** Cascade cycloisomerization/cyclization of yne-enone and enynamide. **C** This work: silver/PPY-catalyzed cycloisomerization/(2 + 3) cycloaddition.

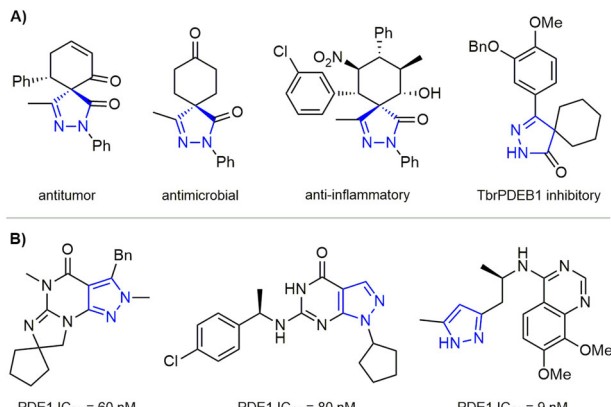

**Fig. 2 Representative bioactive spiro-pyrazolones and pyrazole-containing PDE1 inhibitors. A** Pharmacologically important spiro-pyrazolone scaffolds with druggability. **B** Representative PDE1 inhibitors containing pyrazole pharmacophores.

Spiropyrazolones are scaffolds with great potential for drug discovery due to their diverse biological activities, such as anti-tumor, antimicrobial, and analgesic (Fig. 2A)[101–103]. Furthermore, as the privileged structures, pyrazole and pyrazolone derivatives show good binding abilities to various bioactive proteins. For example, their derivatives have a good inhibitory effect on PDE1 (Fig. 2B)[104–106]. Therefore, it would be interesting to construct a novel spiropyrazolone compound library and study its enzyme binding properties. Our group has a continuing interest in the development of chiral Lewis base catalysts and new strategies for synthesizing medicinally relevant scaffolds[107–111]. Here, we report a relay catalytic system involving silver acetate and a modified chiral PPY, which can efficiently catalyze the unreported cycloi-somerization/(2 + 3) cycloadditions of enynamides (Fig. 1C). As a result, the bispirocyclopentene pyrazolone products, which exhibited promising PDE1B inhibitory activity, could be efficiently synthesized in a highly stereoselective and economical manner.

## Results and discussion

**Optimization of the reaction conditions**. Based on our previous study, we first investigated the reaction of enynamide **1a** with Edaravone-derived MBH carbonate **2a** as the allylic ylide precursor under the dual catalysis of metal and Lewis base catalyst (Table 1, see Tables S1–S3 of the Supplementary Methods for optimization details)[69,112–117]. We found that PPh₃ and DABCO were ineffective, and no reaction occurred. In sharp contrast, using chiral PPY **C1** with PPh₃AuCl (5 mol%) and AgSbF₆ (10 mol%) at 40 °C in chloroform, bispirocyclopentene pyrazolone product **3a** formed from the cycloisomerization/(2 + 3) ylide cycloaddition was obtained in 49% yield with exclusive diastereoselectivity, albeit with moderate enantioselectivity (Table 1, entry 3). In previous reactions, more economical silver salt was usually used as an additive for anion exchange rather than the catalyst for the cycloisomeri-zation of enynamides. Interestingly, in the screening of different metal catalysts, we found AgOTf is a compatible and effective co-catalyst with chiral PPY in the dual catalytic reaction, providing a better yield than the cationic gold catalyst (entry 4). AgOAc, with a lower price, also worked well, affording **3a** in 58% yield with 80:20 er (entry 6).

Encouraged by these results, we evaluated a series of chiral PPY catalysts **C2–C7**. Adjusting the substituents on the phenyl groups of Connon's catalyst **C1** influenced reaction efficiency and enantioselectivity, while only moderate results were obtained. Therefore, we next investigated the bifunctional catalyst **C4–C6** with a C4–OH group on the prolinol ring[69]. Although **C6** could

afford better enantioselectivity, low activity was observed in the reaction. By introducing a steric hindrance group to the *trans*-C4-OH and blocking the H-bond donor, **3a** could be obtained with 85:15 er (**C7**, entry 12). To further improve the stereoselectivity, we designed and prepared two new PPYs (see page S2 of the Supplementary Methods for detailed procedures). **C8** bearing 4-phenyl groups provided **3a** in 86% yield with 86:14 er (entry 13). To our gratification, **C9** with 3,5-diphenyl phenyl groups exhibited better face shielding, delivering higher enantioselectivity (entry 14). Conducting the reaction at 0 °C, the enantioselectivity could be further improved without affecting the diastereoselectivity and efficiency (>19:1 dr, 95:5 er, entry 15).

**Scope of substrates**. Having established the chiral PPY/silver dual catalytic system and optimal reaction conditions, we then focused on the substrate scope for the cycloisomerization/(2 + 3) cycloaddition reaction (see pages S5 of the Supplementary Methods for detailed procedures, Supplementary Data 1 for characterization data, Supplementary Data 2 for NMR and HPLC spectra). First, a range of pyrazolone-derived MBH carbonates **2** was investigated (Fig. 3). Different N-alkyl substituents did not affect the reaction yield and stereoselectivity. N-tert-butyl and N-methyl substituted **2** afforded bispirocyclopentene pyrazolone products **3b** and **3c** in good yield with 97:3 and 91:9 er, respectively. Various **2** with N-aryl groups bearing electron-donating and electron-withdrawing substituents at different positions were well compatible, producing **3d–3i** in 44%–99% yields, with up to 99.5:0.5 er. Changing the methyl group on the pyrazolone ring with the ethyl group led to a slightly decreased yield (**3j**). MBH carbonates **2** bearing an ethyl ester group also worked well, delivering **3k** with good results. Remarkably, the reaction showed exclusive diastereoselectivities in all cases.

Next, the scope of the newly developed catalytic system was further explored by using different enynamides in the reactions with **2a**. Methanesulfonic and 4-nitrobenzenesulfonic enyna-mides **1** are suitable substrates. Corresponding products **3l** and **3m** were obtained in excellent yields with high enantioselec-tivities. All versions of **1** bearing a *para*-, *meta*- and *ortho*-substituent on the phenyl ring of the alkyne moiety were amenable to the conditions (**3n–3s**). The high efficiency, excellent diastereoselectivities, and good enantioselectivities are ascribed to the dual catalytic system and its stereocontrol strategy. Enyna-mides **1** bearing a thiophen-2-yl group was well tolerated (**3t**). Notably, the alkyl group (*t*-butyl and *n*-butyl) substituted enynamides **1** were also compatible in the reaction, albeit with slightly declined enantioselectivities. Then, we tested substrates **1** with different aryl groups on the alkenyl moiety. Similarly, bispirocyclopentene pyrazolone products **3x−3ac** were smoothly obtained with satisfactory results, demonstrating that the electron and steric effect did not affect the reaction much.

**Synthetic application**. To test the practicability of the relay cat-alytic cycloisomerization/(2 + 3) ylide cycloaddition strategy, we conducted a scale-up synthesis. The reaction of **1a** (1.0 mmol) with **2a** afforded 464.9 mg of **3a** as a white powder with good yield and uncompromised stereoselectivity (Fig. 4A, see page S6 of the Supplementary Methods for details). The structure and absolute configuration of bispirocyclopentene pyrazolone **3** were unambiguously identified by X-ray crystallography analysis of **3a** (CCDC 2207417, see Supplementary Data 3 for details). We then explored the transformation of the product (see page S7 of the Supplementary Methods for detailed procedures, Supplementary Data 2 for characterization data, Supplementary Data 3 for NMR and HPLC spectra), which contains a fused cyclopentene, a furan-2(3H)-imine, and a pyrazolone ring. Interestingly, the oxirane-fused polycyclic product **4** could be accessed in 86% yield with

**Table 1 Optimization of the reaction conditions[a].**

| Entry | M | C | Yield (%) | Er |
|---|---|---|---|---|
| 1 | PPh₃AuCl, AgSbF₆ | PPh₃ | N.R. | – |
| 2 | PPh₃AuCl, AgSbF₆ | DABCO | N.R. | – |
| 3 | PPh₃AuCl, AgSbF₆ | C1 | 49 | 76:24 |
| 4 | AgOTf | C1 | 66 | 79:21 |
| 5 | Ag₂CO₃ | C1 | 56 | 72:28 |
| 6 | AgOAc | C1 | 58 | 80:20 |
| 7 | AgOAc | C2 | 26 | 74:26 |
| 8 | AgOAc | C3 | 42 | 70:30 |
| 9 | AgOAc | C4 | 37 | 56:43 |
| 10 | AgOAc | C5 | 66 | 74:26 |
| 11 | AgOAc | C6 | 22 | 83:17 |
| 12 | AgOAc | C7 | 52 | 85:15 |
| 13 | AgOAc | C8 | 86 | 86:14 |
| 14 | AgOAc | C9 | 82 | 89:11 |
| 15[b] | AgOAc | C9 | 80 | 95:5 |

[a]Conditions: **1a** (0.1 mmol) and **M** (10 mol%) in CHCl₃ (1.0 mL) was stirred at 40 °C for 1 h before **2a** (0.1 mmol) and **C** (20 mol%) was added. Then, the mixture was stirred at 40 °C for 3 h.
[b]After **2a** and **C9** were added, the reaction was stirred at 0 °C for 48 h and then at 40 °C for 8 h.

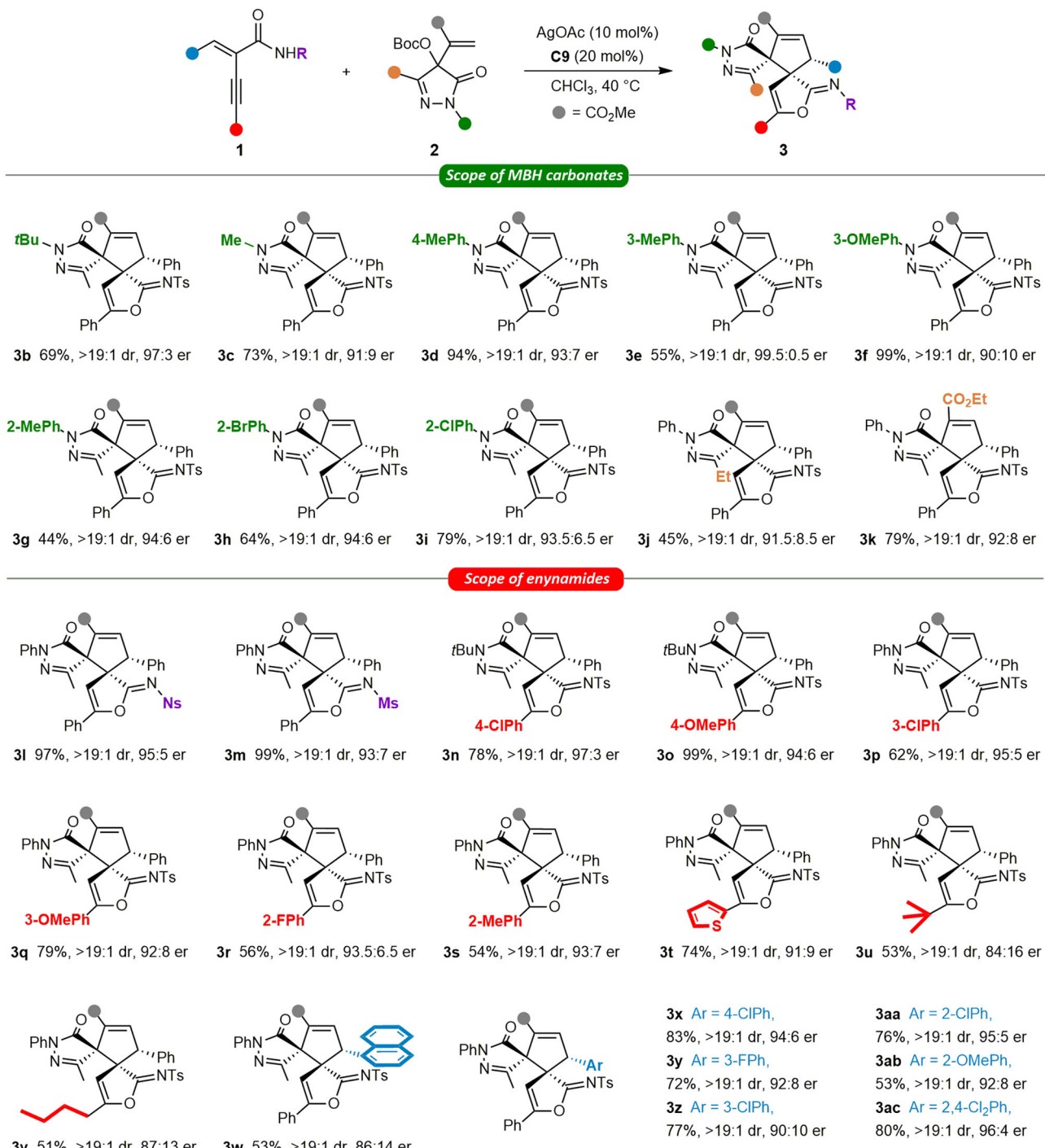

**Fig. 3 Substrate scope of the MBH carbonates and enynamides[a].** [a]Conditions: **1** (0.1 mmol) and AgOAc (10 mol%) in CHCl₃ (1.0 mL) was stirred at 40 °C for 1 h. Then, **2** (0.1 mmol) and **C9** (20 mol%) were added, and the reaction was stirred at 0 °C for 48 h and at 40 °C for another 3 h.

excellent diastereoselectivity by treating **3a** with *meta*-chloroperoxybenzoic acid (>19:1 dr, Fig. 4B). Remarkably, the switchable divergent synthesis of different isomers was achieved by treating **3a** with acid or base, delivering diastereomer **3a'** and isomer **5** in good yield with exclusive diastereoselectivity (Fig. 4C). The spiropyrazolone could serve as a directing group, allowing efficient late-stage modification of **3a** through C–H functionalization. Pd-catalyzed acyloxylation and Rh-catalyzed coupling with diazo esters delivered products **6** and **7** in high yields with exclusive regioselectivity, respectively.

**Control experiments**. To provide insights into the reaction mechanism, we conducted some control experiments (Fig. 5A, see page S5 of the Supplementary Methods for details). First, we investigated the formation of furan-based azadiene **A**. Using 10 mol% of silver acetate as a single catalyst or adding both chiral PPY **C9**/silver acetate led to very similar results. Azadiene intermediate (*E*)-**A** was efficiently generated as the major product with a trace amount of *iso*-**A**, showing that PPY did not influence the cycloisomerization step. Next, control experiments were conducted to disclose the catalyst effect of the subsequent (2 + 3) ylide

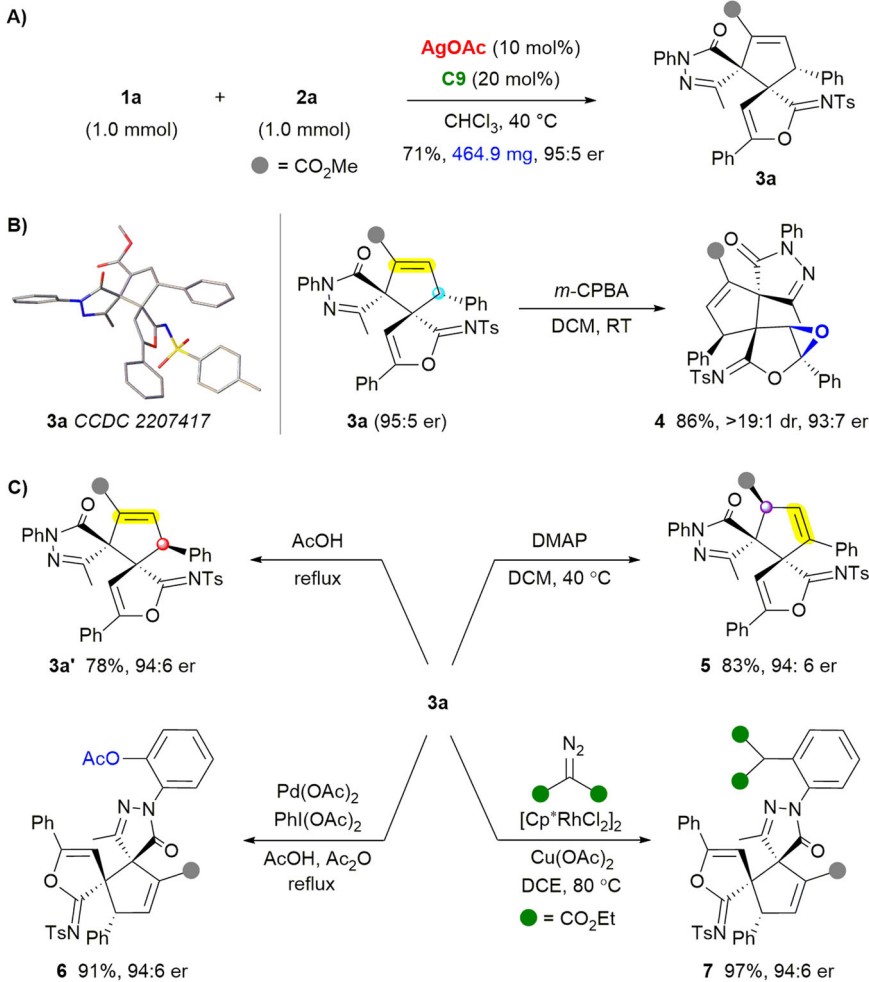

**Fig. 4 Scale-up synthesis and transformations of product 3a. A** Scale-up synthesis of **3a**. **B** Stereoselective epoxidation. **C** Stereodivrgent isomerization and site-selective C-H functionalization.

cycloaddition. By treating (*E*)-**A** with **C9** as the single catalyst, the reaction gave **3a** in 85% yield with 95:5 er. The conversion and stereoselectivity are similar to those obtained by dual catalysis, indicating that the reaction may undergo a silver/PPY relay catalytic process. AgOAc and chiral PPY showed good compatibility in the reaction, which is essential to the strategy. Therefore, we further studied the catalytic efficiency when the loading of **C9** was reduced to 15 mol% and 10 mol%. Product **3a** could be obtained in 67% and 45% yields, respectively, and the stereoselectivity was unaffected. These results showed that even an equimolar amount of AgOAc would not deactivate the Lewis base.

**DFT computational calculations and proposed mechanism.** Then, density functional theory (DFT) computational calculations were conducted to rationalize the chemoselectivity in this silver/chiral PPY relay catalytic reaction (see Supplementary Data 4 for details). The chemoselectivity is determined in the PPY-catalyzed cycloaddition step, in which (4 + 3) cycloaddition would give the fused-furan product, and the (2 + 3) cycloaddition would deliver the spirocyclic furan-2(3*H*)-imine product **3**. The optimized geometries of the key transition state **TS1** and **TS2** are given in Fig. 5B. The relevant computational details and cartesian coordinates of optimized structures are provided in Supplementary Data 4. Comprising energies of **TS1** and **TS2** reveal that the (2 + 3) cycloaddition is more favored than the (4 + 3) cycloaddition by 19.4 kcal mol⁻¹. It may originate from the steric

hindrance between the phenyl group of **1a** and the *N*-Ts group in the structure of **TS2**. Moreover, natural population analysis (NPA) found that the electron deficiency at the internal C2 position is more significant than that of the terminal *N*-atom. Accordingly, a relay catalytic mechanism involving Lewis acid-catalyzed cycloisomerization and chiral PPY-catalyzed asymmetric (2 + 3) ylide cycloaddition was proposed (Fig. 6).

**PDE1 inhibition of product 3.** Considering the potential inhibitory effect of the known pyrazolone and pyrazole derivatives on PDE, we next evaluated the binding abilities of the synthesized bispirocyclopentene pyrazolones to PDE1. As depicted in Fig. 7A, we investigated the PDE1B inhibitory capacities of **3** at a concentration of 0.1 μmol·L⁻¹. To our gratification, **3a** exhibited a good inhibitory ratio. Interestingly, the inhibitory activities of **3b** and **3c** declined with *N*-alkyl substitution on the pyrazolone fragment. For products containing various *N*-phenyl groups bearing electron-donating and electron-withdrawing substituents at different positions, the PDE1B inhibition ratios were not significantly affected (**3d–3i**). Replacing the methyl group on the pyrazolone ring with the ethyl group led to the loss of inhibitory activity, possibly due to the steric hindrance (**3j**). The change of the methyl ester on the cyclopentene ring to the ethyl ester displayed a marginal effect on the activity (**3k**). For products from different enynamides **1**, the methanesulfonamide or 4-nitrobenzene-sulfonamide substitution significantly weakened

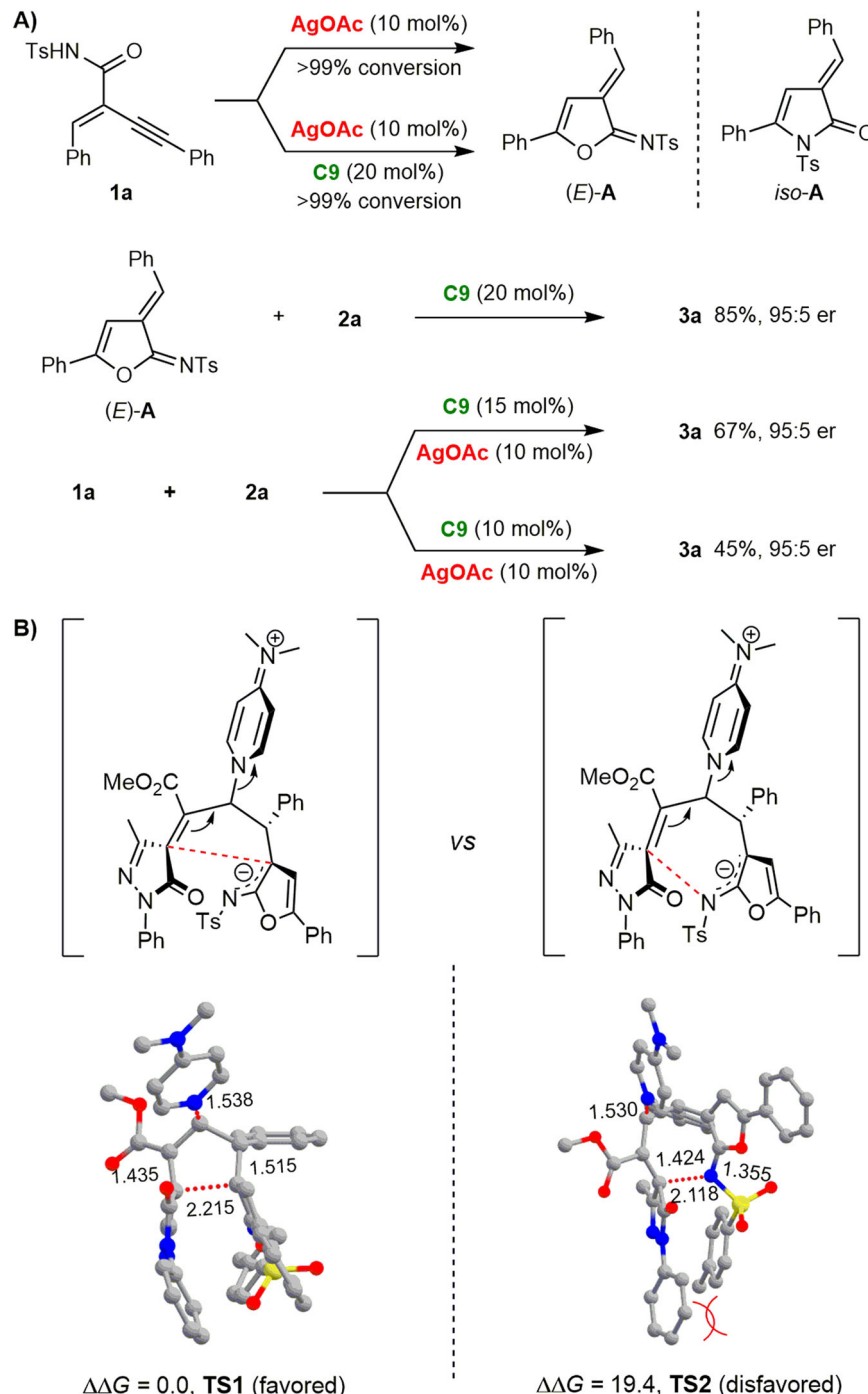

**Fig. 5 Control experiments and computational calculations. A** Catalyst effect studies. **B** DFT-optimized structures and relative free energies ($\Delta G$, kcal/mol) of transition states **TS1** and **TS2** along two possible chemoselective pathways at the M06/Def2-TZVP-SMD(CHCl$_3$)//M06/6-31 G(d,p)-SMD(CHCl$_3$) level.

the inhibition (**3l** and **3m**). **3n–3q** bearing a *para-* or *meta-*substituent on the phenyl ring exhibited a good inhibitory ratio. In contrast, other derivatives **3** from enynamides bearing *ortho-*substituted phenyl, alkyl and thienyl alkyne moiety have reduced activities (**3r–3v**). Notably, enynamides **1** bearing different substituted aryl groups on the alkenyl moiety could provide **3** with enhanced inhibitory activities (**3x–3ac**), especially compound **3x** with a 4-Cl-phenyl group.

Then, the IC$_{50}$ values of three PDE1 subtypes were evaluated by screening the compounds whose inhibition ratio of PDE1B exceeded 60% at 0.1 μM (Fig. 7B). These compounds generally

had an excellent inhibitory effect on PDE1B, and there was no significant specificity among the three PDE1 subtypes. The subtype selectivity of compound **3x** across the PDE family was determined and listed in Table S4. It only showed weak inhibitory activities on PDE4 and PDE5 and low inhibitory activities on other PDE subtypes (Fig. 7C). The interaction modes of **3x** on PDE1B and PDE3 were studied by molecular docking and molecular dynamics simulations. Figure 7D, E shows the 3D and 2D contour of the binding conformation of **3x** and PDE1B after a 100-ns scale molecular dynamics simulation. The RMSD curves of PDE1B or PDE3A with or without compound **3x** were depicted

**Fig. 6 Proposed mechanism for the silver/chiral PPY relay catalytic cycloisomerization/(2 + 3) cycloaddition cascade reaction of enynamide.** The Ag-catalyzed cycloisomerization of enynamide **1a** generates the azadiene intermediate (*E*)-**A**, while chiral PPY catalyst **C9** converts MBH carbonate **2a** into the allylic ylide intermediate **C**. This intermediate then undergoes the (2 + 3) cycloaddition with (*E*)-**A** to produce bispirocyclopentene product **3a**.

in Fig. 7F, and the RMSD values of two **3x**-protein complexes were relatively stable during the simulation time, which suggested that compound **3x** rapidly reached the equilibrium state in the binding pocket of PDEs. The hydrogen bond between the ester group and Ser272, π-π stacking with Phe424, and hydrophobic interactions with Met389, Leu409, and Val417 contributed to the selective binding of **3x** to PDE1B (Fig. 7G).

**The therapeutic effects and preliminary mechanism of compound 3x.** According to the important role of PDEs in inflammatory pulmonary damage and fibrosis, the therapeutic effects and preliminary mechanism of compound **3x** were evaluated both in vitro and in vivo (Fig. 8). The expression levels of fibrosis markers, such as Fibronectin, Collagen-I, and α-SMA, were stimulated by TGF-β1 and suppressed after **3x** incubation (Fig. 8A). Similar results were observed by the immunofluorescence staining of fibronectin and α-SMA in vitro (Fig. 8B, C). On the bleomycin (BLM)-induced idiopathic pulmonary fibrosis (IPF) rat model, **3x** demonstrated a comparable anti-fibrosis effect to the clinically approved drug pirfenidone (PFD, Fig. 8D, E). The pulmonary ventilation markers, such as end-inspiratory pause (EIP), end-expiratory pause (EEP), mid-expiratory flow (EF50), peak inspiratory flow (PIF), and peak expiratory flow (PEF) were alleviated after **3x** or PFD treatment in BLM-induced IPF rat models, suggested that the therapeutic capacity of **3x** on IPF related respiratory dysfunction. The morphological changes of lung tissues were checked by hematoxylin-eosin (H&E) staining or Masson's trichrome staining of collagen deposition. Compared with the control group, remarkable morphological changes and collagen deposition on the alveolar bronchi were observed after BLM administration, and the treatment of PFD or **3x** significantly reduced these pathological changes. Both in vitro and in vivo results suggested that **3x** could ameliorate BLM-induced pulmonary fibrosis *via* inhibiting PDE1.

## Conclusion

In conclusion, we have developed a metal/Lewis base relay catalytic system, featuring silver acetate and a chiral PPY based on chiral 4-hydroxy diarylprolinol. The catalytic system was successfully applied in the cycloisomerization/(2 + 3) cycloaddition reaction of enynamides, which has not been realized before. Bispirocyclopentene pyrazolone products could be efficiently synthesized

in a highly stereoselective and economical manner. Moreover, simple transformations of the product could access stereodivergent diastereoisomers and densely functionalized polycyclic derivatives. Control experiments and DFT calculations illustrated the relay catalytic model and the origin of the uncommon chemoselectivity. In subsequent bioassays, the products containing a privileged drug-like scaffold exhibited isoform-selective PDE1 inhibitory activity in vitro. Compound **3x** displayed a good therapeutic effect for ameliorating BLM-induced pulmonary fibrosis *via* inhibiting PDE1 in vivo. We expect this powerful catalytic system to be widely applied in stereoselective construction of other valuable molecules in the future.

## Methods

**General procedure for the cycloisomerization/(2 + 3) cycloaddition reaction.** A mixture of enynamides **1** (0.10 mmol), AgOAc (1.7 mg, 0.01 mmol, 10 mol%) in CHCl₃ (0.5 mL) was stirred at 40 °C for 1 h, MBH carbonate **2** (0.10 mmol), **C9** (19.7 mg, 0.02 mmol, 20 mol%) in CHCl₃ (0.5 mL) were added to the above solution and stirred at 0 °C for 48 h, and at 40 °C for another 3 h until the reaction was complete (determined by TLC analysis). The mixture was concentrated under vacuum and purified by column chromatography on silica gel (petroleum ether: ethyl acetate: dichloromethane = 10:1:1 to 5:1:1) to afford the pure products **3**.

**Bioassay of phosphodiesterase PDE1 and other PDE subfamilies.** The PDE1B protein was purified according to the protocols described in previous report. PDE activity was measured by a scintillation proximity assay using a fixed amount of enzyme and substrate concentrations. The phosphodiesterase (PDE) assays measure the conversion of H³-cAMP for PDE 1A, 1B, 1C, 3A1, 4D2, 7A2, 8A2 and 10A1) or H³-cGMP for PDE 2A, 5A1, 6C, 9A2 and 11A4, by the relevant PDE enzyme subtype. The scintillation proximity beads bind selectively to H³-AMP or H³-GMP, with the magnitude of radioactive counts being directly related to PDE enzymatic activity. In brief, 1 μL of test compound in dimethyl sulfoxide was added to each well. Enzyme solution was then added to each well in buffer (Trizma and MgCl₂) containing Brij 35 (0.01% (v/v)). For PDE1 subtype assays the buffer additionally included CaCl₂ (30 mM) and calmodulin (25 U ml⁻¹). Subsequently, 20 μL of H³-cGMP (or 20 μL of H³-cAMP) was added to each well to start the reaction and the plate was incubated for 30 min at 25 °C. Following an additional 8 h incubation period the plates were read on a MicroBeta radioactive plate counter to determine radioactive counts per well.

**Computational procedures in molecular dynamics simulations.** For each system, energy minimization and MD simulation were performed by using the Gromacs-2020.6 package. The AMBER99 and GAFF forcefield were utilized to build the topology of protein and ligand molecules, respectively. Prior to MD simulations, the entire system was subject to energy minimization in two stages to remove bad contacts between the complex and the solvent molecules. Firstly, the water molecules and counterions were minimized by freezing the solute using a

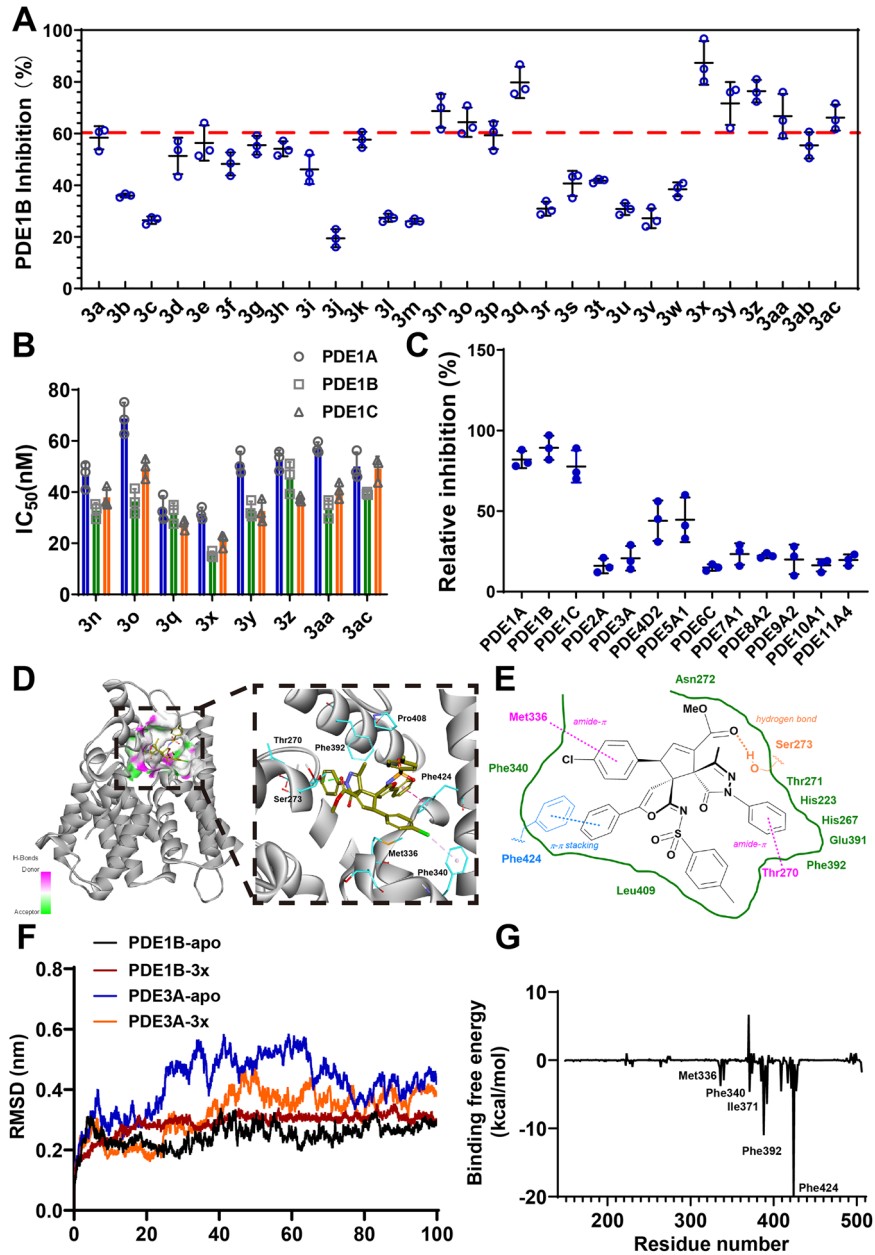

**Fig. 7 PDE1 inhibition of product 3. A** The inhibitory rate of compounds **3** on PDEIB @0.1 μmol·L$^{-1}$. **B** In vitro PDE1 inhibition of selective compounds on three PDE1 subtypes. **C** Relative inhibition of **3x** on the PDE family proteins @0.1 μmol·L$^{-1}$. **D** 3D contour of the binding conformation of **3x** and PDE1B. **E** 2D contour of the binding conformation of **3x** and PDE1B. **F** The RMSD (root-mean-square deviations) of the protein main chain atoms compared to the initial conformer. **G** Decomposition of the individual component of binding free energies of **3x**-PDE1B complex by MM/PBSA. The error bars indicated the standard errors with mean values in each group (Mean ± SD).

harmonic constraint of a strength of 100 kcal mol$^{-1}$Å$^{-2}$. Secondly, the entire system was minimized without restriction. Each stage was consisted of a 5000-step steepest descent and a 5000-step conjugate gradient minimization. In MD simulations, Particle Mesh Ewald (PME) was employed to deal with the long-range electrostatic interactions. The cutoff distances for the long-range electrostatic and van der Waals energy interaction were set to 10 Å. THE SHAKE procedure was utilized, and the time step was set to 2 fs. The systems were gradually heated in the NVT ensemble from 0 to 300 K over 500 ps and equilibrium in the NPT ensemble over 500 ps. Then, 100-ns scale MD simulations were performed under the constant temperature of 300 K. During the sampling process, the coordinates were saved every 10 ps and the conformations generated from the simulations were used for further binding free energy calculations and decomposition analysis.

**Cell culture and western blotting.** The HLF cells were purchased from Procell Life Science Technology and was cultured in F12K with 10% fetal bovine serum (FBS) and 1% penicillin/streptomycin (both from Gibco; Thermo Fisher Scientific,

Inc.). All the cell lines were maintained at 37 °C in a humidified incubator with 5% CO$_2$. The cells were harvested and lysed with RIPA (Beyotime Institute of Biotechnology). The protein concentration of each sample was measured using a Pierce™ Rapid Gold BCA Protein Assay kit (Thermo Fisher Scientific, Inc.) based on the manufacturer's guidelines. Total protein was separated using 12.5% SDS-PAGE, transferred to PVDF membranes, blocked with 5% skimmed milk at room temperature for 2 h, then incubated with the following primary antibodies on a shaker overnight at 4 °C. Following which, the membranes were washed with TBS containing 0.1% Tween-20 three times and incubated with HRP-conjugated secondary antibodies (1:10,000 dilution; ProteinTech Group, Inc.) for 1 h at room temperature. The blotted proteins were observed using Immobilon ECL Ultra Western HRP Substrate (Merck KGaA), scanned with a Chemi-Doc System (Bio-Rad Laboratories, Inc.) and analyzed using ImageJ software (https://imagej.net).

**Immunofluorescence (IF) assays.** In total, about $1 \times 10^5$ HLF cells administrated with TGFβ1 and/or compound **3x** were plated on coverslips, cultured overnight at

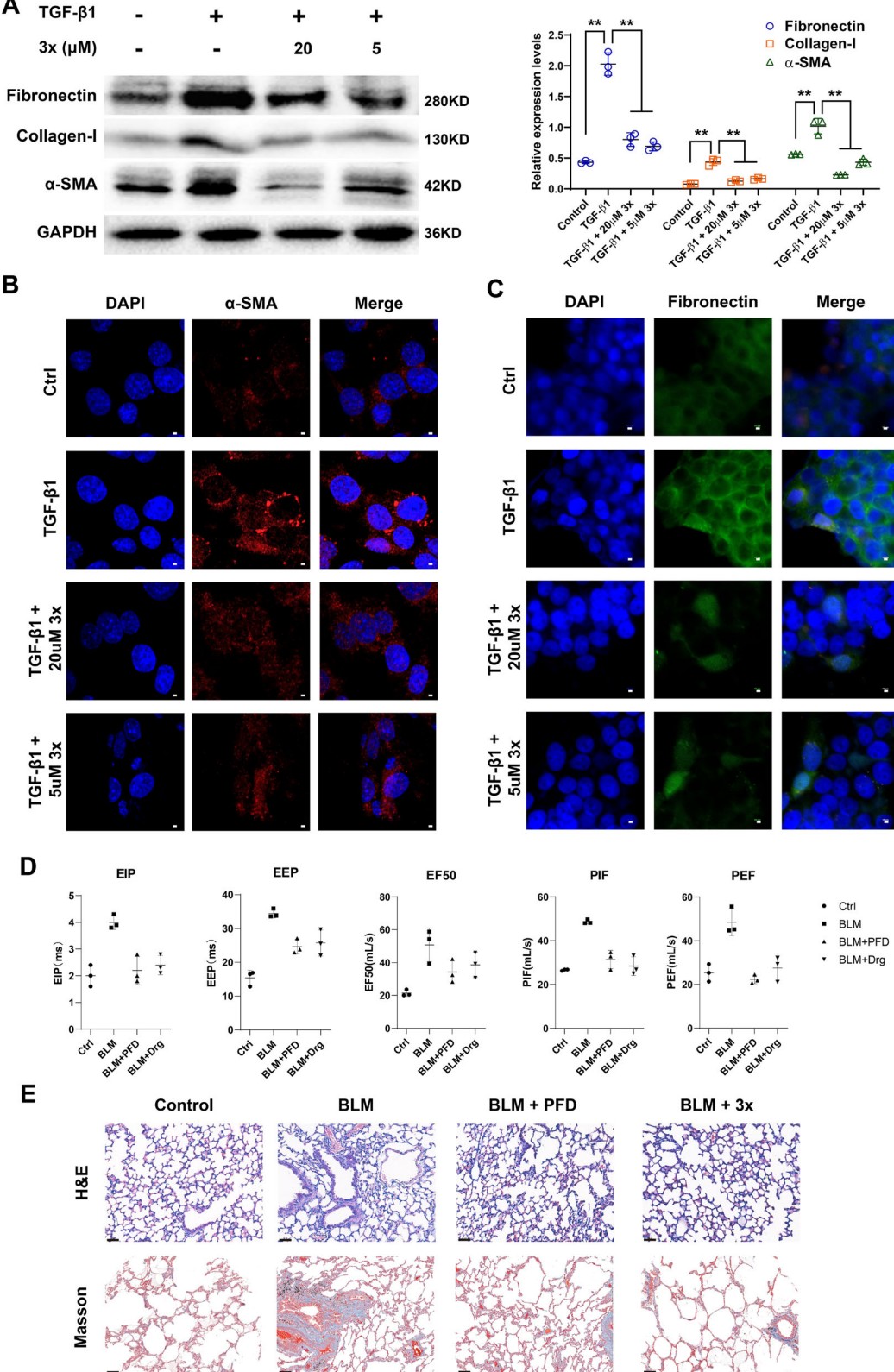

**Fig. 8 The therapeutic effects and preliminary mechanism of compound 3x. A** Western blot analysis of Fibronectin, Collagen-I, α-SMA and GAPDH expression levels. (**$p < 0.01$; student's $t$-test). **B** Immunofluorescence staining analysis for the expression of α-SMA proteins, Scale bar = 10 μm. **C** Immunofluorescence staining analysis for the expression of Fibronectin, Scale bar = 10 μm. **D** The bar graph shows the pulmonary respiratory function of different groups of rats. **E** Representative hematoxylin-eosin (H&E) staining sections and Masson staining sections from the pulmonary tissues of the control, BLM-treated, BLM + PFD-treated, and BLM + **3x**-treated groups, Scale bar = 50 μm. The error bars indicated the standard errors with mean values in each group (Mean ± SD).

37 °C, the coverslips were fixed with 4% pro-cooled paraformaldehyde for 20 min at room temperature, these cells were fixed in 3.7% formalin (Sigma-Aldrich), permeabilized in 0.25% Triton X-100 (Sigma-Aldrich), and blocked with 10% goat serum for 1 h. After overnight incubation with the primary antibody diluted with 10% goat serum, we added 250 μL of the fluorescent secondary antibody solution (1:100, diluted with 10% goat serum) and incubated at room temperature for 1 h in the dark, then briefly incubated with DAPI (Invitrogen; Thermo Fisher Scientific, Inc.) at room temperature for 5 min in the dark. Finally, the slides were sealed with neutral balsam and viewed using a confocal fluorescence microscope (Axiovert 200 M; Zeiss GmbH).

**BLM-induced pulmonary damage rat model**. All relevant animal care and experimental protocols were in accordance with the "Guide for the Care and Use of Laboratory Animals" (National Institutes of Health Publication, revised 1996, No. 86-23, Bethesda, MD) and approved by the Institutional Ethical Committee for Animal Research of Chengdu University of Traditional Chinese Medicine (No. 2022-37). After a habituation period of 1 week, the animals were randomly assigned into four groups: control group, model group, **3x** (20 mg kg$^{-1}$) group and positive control group (PFD 150 mg kg$^{-1}$). The modeling method was implemented in the model and the **3x** group as follows: after the rats were anesthetized by an intraperitoneal injection of 4% pentobarbital sodium (10 mL kg$^{-1}$), the lower neck was incised aseptically, dissected bluntly, and well exposed; then, about 0.2 mL of bleomycin (5 mg kg$^{-1}$) was injected and the rats were immediately erected and rotated several times to make the liquid distribute evenly. After the wounds were sutured, states of the rats after recovery were observed. In the meantime, the rats in the control group were injected with the same amount of normal saline into the trachea. After 28 days of administration, the respiratory level in each group was measured. Then, the rats were anesthetized by an intraperitoneal injection of 4% pentobarbital sodium, and left lower pulmonary lobes were harvested after the rats were euthanized; the tissues were immersed in 4% buffered paraformaldehyde at room temperature overnight and then embedded in paraffin wax. Pulmonary samples were stained by the H&E or Masson's trichrome staining. An Olympus FV-3000 microscope was used to examine the stained pulmonary sections.

**Reporting summary**. Further information on research design is available in the Nature Portfolio Reporting Summary linked to this article.

## Data availability

Detailed experimental details are available in the Supplementary Methods. Full characterization data of compounds can be found in Supplementary Data 1. $^1$H, $^{13}$C NMR, $^{19}$F NMF spectra, and HPLC chromatograms can be found in Supplementary Data 2. The X-ray crystallographic coordinates for structures reported in this Article have been deposited at the Cambridge Crystallographic Data Centre (CCDC), under deposition number CCDC 2207417 (**3a**). These data can be obtained with reference to Supplementary Data 3 or free of charge from The Cambridge Crystallographic Data Centre via www.ccdc.cam.ac.uk/data_request/cif. Computational chemistry details are available in Supplementary Data 4. The experimental procedures for the bioassay and uncropped images from western blots can be found in Supplementary Data 5.

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

## Acknowledgements

We are grateful for financial support from the National Natural Science Foundation of China (Nos. 82073998 and 22001024), the Science & Technology Department of Sichuan Province (Nos. 2022JDRC0045), Innovation Team and Talents Cultivation Program of National Administration of Traditional Chinese Medicine (No. ZYYCXTD-D-202209).

## Author contributions

J.J. undertook most of the experimental work and analytical characterization. J.Z. and Y.L. synthesized some substrates. G.H. conducted the bioactivity evaluation. W.H. and C.P. guided part of this study. G.Z. and B.H. initiated the project, wrote the manuscript and supervised the projects.

## Competing interests

The authors declare no competing interests.
