## [Peer Review File · Communications Chemistry]

Reviewers' comments:

Reviewer #1 (Remarks to the Author):

In this manuscript, the authors reported cycloisomerization/(2+3) cycloaddition reaction of enynamides with pyrazolone-derived MBH carbonates through a metal/Lewis base relay catalytic system featuring silver acetate and a modified chiral pyrrolidinopyridine. Bispriocyclopentenepyrazolone products were produced in acceptable yields and enantioselectivities. In principle, the current work extended the reaction partner of pyrazolone-derived MBH carbonates to include a type of conjugate imine formed in situ whose carbon-carbon double bond was involved in the sequential reaction. Therefore, the novelty of the current work is moderate. However, the work has been thoroughly studied and provided the useful information for research on reactions of pyrazolone-derived MBH carbonates. The reviewer recommends accepting the manuscript for publication in Communications Chemistry after revision of the following points.

1. HRMS of the compound 3g is wrong.
2. NMR spectra of the compound 6 is not qualified. Please remove the solvent residue in the compound 6.

Reviewer #2 (Remarks to the Author):

Review report

1)The advantage of the present catalytic systems and resolution of unsolved issues

As illustrated in Figure 1A, there have been no reports on the dual catalysis comprised of metal/chiral 4-aminopyridine derivatives. The authors developed their original PPY catalyst (C9) could work in the presence of silver salt (Table 1). The effect of substituents of catalysts (C1-C9) on the chemical yields of 3a is impressive (Table 1, entries 6-15), but there seems to be no conclusion or description on the effect. The bulky OTBDPS group and/or bis-(3,5-diphenylphenyl) group would prevent to coordinate of the pyridine to the metal center. Such information is beneficial for the readers developing metal catalysis as well as the organocatalysis. As illustrated in Figure 1B, cycloisomerization-[2+n] cycloaddition reaction has not been reported. However, it is hard to understand the differences between cycloisomerization-[3+n] or cycloisomerization-[4+n]. It would be better to add the type of products, such as e.g. spirocycles versus fused-bicycles.

2)The characterization of the products

All new compounds were well-characterized in the supplementary information. ¹H and ¹³C NMR as well as the HPLC chart seems to be enough fine. Some of the product structures (S23-) are overlapped with the data, and this error should be solved.

3)References

Overall, the appropriate literature citation has been made.

4)The proposed reaction mechanism

The authors provided two possible chemoselective pathways in Figure 4B. This reviewer is not sure why TS2 is compared with TS1. Has the 7-membered byproducts been isolated? Rather, it is recommended to provide the TS for the minor enantiomer, and how the chiral catalyst functions in the cyclization step. In related to this issue, this reviewer is wondering if the 1,4-addition step (Figure 5, int C to int D) is reversible or not. In addition, why (E)-A was favored than the (Z)-isomer in the silver-catalyzed cyclization step (Figure 5)? Overall, Figure 5 is not so kind for readers to understand the mechanism, and e.g. the coordination of alkyne to silver and the additional curly arrows help to understand.

5)Additional point

A) Which kind of interaction is supposed for the highly enantioselective production of compound 3b (94%ee), because N-t-Bu group could not have pi-pi interaction as similarly as 3a (Figure 4A).

B) Is it possible to convert NTs group into other groups? If possible, accessible compounds classes will be expanded, and leads to further SAR studies.

As described above, this manuscript contains enough novelty and the overall chemistry seems very sound. For these reasons, this reviewer is positive for publication of this manuscript in Communications Chemistry, after the above-mentioned points are corrected or mentioned.

Reviewer #3 (Remarks to the Author):

This manuscript by Han and coworkers reported a silver acetate and modified chiral PPY relay catalytic cycloisomerization/(2 + 3) cycloaddition reaction of enynamides, giving various chiral bispirocyclopentene pyrazolone derivatives, which further exhibited isoform-selective PDE1 inhibitory activity. This work is meaningful and deserves publication in Commun. Chem. after addressing the following problems.

1. In the introduction, the authors described the significance of developing new multicatalytic systems and mentioned that dual catalytic system of 4-aminopyridine with transition metal is still untapped. Actually, some related works have been reported (Chen, *Angew. Chem. Int. Ed.* 2019, 58, 15021-15025; Lu, *Angew. Chem. Int. Ed.* 2021, 60, 1845-1852), which should be added to the references.

2. In the optimization of the reaction conditions, did the authors try to further change the AgOAc loading? In the ESI, the table S2 shows the effect of solvents on this reaction. Can the authors explain the necessity of using solvent 1, which would be removed before adding substrate 2a and catalyst C9.

3. The substrate scope of the reaction is broad and sufficient. A range of MBH carbonates 2 were investigated by varying the substituents. However, only ethyl group substituted substrate at the C3 position of the pyrazolone moiety was examined, and what is the result of the reaction with an aryl group? In addition, the diastereoselectivities of all products should be provided in Table 2.

The transformations of product 3a were also explored in this paper. How the authors explain the slight increase of enantioselectivity after formation of product 6 (from 95:5 er to 97:3 er). It seems to be weird because the stereogenic center, in general, is stable and hard to change.

Comments from reviewer 1:

1) **Comment:** *In this manuscript, the authors reported cycloisomerization/(2+3) cycloaddition reaction of enynamides with pyrazolone-derived MBH carbonates through a metal/Lewis base relay catalytic system featuring silver acetate and a modified chiral pyrrolidinopyridine. Bispriocyclopentenepyrazolone products were produced in acceptable yields and enantioselectivities. In principle, the current work extended the reaction partner of pyrazolone-derived MBH carbonates to include a type of conjugate imine formed in situ whose carbon-carbon double bond was involved in the sequential reaction. Therefore, the novelty of the current work is moderate. However, the work has been thoroughly studied and provided the useful information for research on reactions of pyrazolone-derived MBH carbonates. The reviewer recommends accepting the manuscript for publication in Communications Chemistry after revision of the following points.*

Response: We sincerely thank Reviewer 1 for the thorough review of our manuscript and their valuable feedback. We agree that our work represents an extension of the asymmetric (3 + 2) cyclization reaction of MBH carbonate derived from pyrazolone, but we respectfully disagree that the novelty of our work is only moderate. We believe that our study makes several noteworthy contributions.

Firstly, we have established a dual catalytic system consisting of silver acetate and a modified chiral PPY Lewis base catalyst, which has not been reported before. This represents a significant innovation in the development of efficient and selective catalytic systems. The good compatibility has implications for the design of future reactions using similar Au/Ag-chiral aminopyridine dual catalytic systems, which are undergoing in our group now. Secondly, we demonstrated that the intermediates from cycloisomerization of enynamides could undergo a reaction pathway with different chemoselectivity, which provides new access to spirocyclic products. Finally, we have discovered the interesting PDE1 inhibitory activity of the spiropyrazolone products synthesized in the reaction. This represents a potentially important application of our synthetic methodology. Overall, we believe that our study makes a valuable

contribution to the field of chemistry and provides useful information for future research on the development of efficient and selective catalytic systems.

2) **Comment:** *1. HRMS of the compound 3g is wrong. 2. NMR spectra of the compound 6 is not qualified. Please remove the solvent residue in the compound 6.*

Response: We check the raw HRMS data. The typo in the HRMS data of **3g** has been corrected. We repeated the reaction to get a clean product **6** without solvent residue. The NMR spectra were updated in the SI (see on the next page). Once again, we appreciate Reviewer 1's valuable feedback and the time and effort that they have invested in reviewing our manuscript.

Comments from reviewer 2:

1) **Comment:** *1) The advantage of the present catalytic systems and resolution of unsolved issues*

As illustrated in Figure 1A, there have been no reports on the dual catalysis comprised of metal/chiral 4-aminopyridine derivatives. The authors developed their original PPY catalyst (C9) could work in the presence of silver salt (Table 1). The effect of substituents of catalysts (C1-C9) on the chemical yields of 3a is impressive (Table 1, entries 6-15), but there seems to be no conclusion or description on the effect. The bulky OTBDPS group and/or bis-(3,5-diphenylphenyl) group would prevent to coordinate of the pyridine to the metal center. Such information is beneficial for the readers developing metal catalysis as well as the organocatalysis. As illustrated in Figure 1B, cycloisomerization-[2+n] cycloaddition reaction has not been reported. However, it is hard to understand the differences between cycloisomerization-[3+n] or cycloisomerization-[4+n]. It would be better to add the type of products, such as e.g. spirocycles versus fused-bicycles.

Response: We appreciate Reviewer 2's insightful comments and suggestions. We have carefully considered their feedback and have made revisions to improve the clarity of our manuscript.

Indeed, we observed that the substituents on the diphenyl rings and on the C4-OH group of the prolinol ring would influence the reaction efficiency and enantioselectivity. However, this effect is more complicated and seems to be caused by a combination of factors, not just the steric hindrance effect of the substituent. Therefore, we did not draw an overall conclusion.

Regarding the differences between cycloisomerization-[3 + n] and cycloisomerization-[4 + n], we used different colors in both reactions and specified the reactants and products of the [3 + n] reaction to make it easier for readers to understand. Additionally, we have added a reference below the reaction to provide more context and support for the reader.

2) **Comment:** *2) The characterization of the products*

All new compounds were well-characterized in the supplementary information. ¹H and ¹³C NMR as well as the HPLC chart seems to be enough fine. Some of the product structures (S23-) are overlapped with the data, and this error should be solved.

Response: We sincerely thank Reviewer 1 for the thorough review of our manuscript, supplementary information, and valuable feedback. We have corrected the overlap error on page S23 and have carefully reviewed the spectra part.

3) **Comment:** *3) References: Overall, the appropriate literature citation has been made.*

4) *The proposed reaction mechanism*

The authors provided two possible chemoselective pathways in Figure 4B. This reviewer is not sure why TS2 is compared with TS1. Has the 7-membered byproducts been isolated? Rather, it is recommended to provide the TS for the minor enantiomer, and how the chiral catalyst functions in the cyclization step. In related to this issue, this reviewer is wondering if the 1,4-addition step (Figure 5, int C to int D) is reversible or not.

Response: We sincerely appreciate the thoughtful comments from reviewer 2. The observed chemoselectivity in our work is indeed intriguing and has not been reported previously. We did not detect any (4 + 3) byproduct, and the reaction exclusively provided the (2 + 3) product. This is surprising because aza-diene intermediates obtained from cycloisomerization typically participate in (4 + n) cyclization reactions to yield a variety of fused furans. In order to provide a possible explanation for the chemoselectivity, we compared TS1 with TS2. While we agree that it would be interesting to investigate the stereocontrol of the chiral catalyst in the cyclization step, our focus in this work is to discuss the chemoselectivity. Due to the relatively large size of the modified chiral catalyst and the need to study multiple conformations, the computational workload would be very high. Therefore, we have planned to reveal the mechanism of stereocontrol in this type of catalyst in future work.

4) **Comment:** *In addition, why (E)-A was favored than the (Z)-isomer in the silver-catalyzed cyclization step (Figure 5)? Overall, Figure 5 is not so kind for readers to*

understand the mechanism, and e.g. the coordination of alkyne to silver and the additional curly arrows help to understand.

Response: We sincerely thank reviewer 2 for the comments and suggestions. We have made revisions to Figure 5 to make it more comprehensible. Regarding the aza-diene intermediate, only (*E*)-**A** and a trace amount of *iso*-**A** were generated under the reaction condition. No (*Z*)-isomer of intermediate **A** was observed.

5) **Comment:** A) Which kind of interaction is supposed for the highly enantioselective production of compound **3b** (94%*ee*), because *N*-*t*-Bu group could not have *pai-pai* interaction as similarly as **3a** (Figure 4A).

B) Is it possible to convert NTs group into other groups? If possible, accessible compounds classes will be expanded, and leads to further SAR studies.

Response: We appreciate the valuable comment from reviewer 2. As presented in Table 2, products **3b**, **3c**, **3n**, and **3o** were obtained with high enantioselectivity, suggesting that the π - π interaction did not play a role in stereocontrol. We speculate that the bulky chiral catalyst could induce stereochemical recognition by shielding one face of the ylide intermediate, thus leading to the highly enantioselective production of compound **3**. Besides the *N*-Ts substituted enynamides, we also tested the reaction of *N*-Ph enynamide. However, no aza-diene intermediate was generated under the reaction conditions.

Comments from reviewer 3:

1) **Comment:** This manuscript by Han and coworkers reported a silver acetate and modified chiral PPY relay catalytic cycloisomerization/(2 + 3) cycloaddition reaction of enynamides, giving various chiral bispirocyclopentene pyrazolone derivatives, which further exhibited isoform-selective PDE1 inhibitory activity. This work is meaningful and deserves publication in *Commun. Chem.* after addressing the following problems.

1. *In the introduction, the authors described the significance of developing new multicatalytic systems and mentioned that dual catalytic system of 4-aminopyridine with transition metal is still untapped. Actually, some related works have been reported (Chen, Angew. Chem. Int. Ed. 2019, 58, 15021-15025; Lu, Angew. Chem. Int. Ed. 2021, 60, 1845-1852), which should be added to the references.*

Response: We sincerely appreciate reviewer 3 for bringing to our attention the related works that we have missed in our references. We have carefully reviewed the two works (Chen, Angew. Chem. Int. Ed. 2019, 58, 15021-15025; Lu, Angew. Chem. Int. Ed. 2021, 60, 1845-1852) and added them to the revised manuscript. Thank you for helping us improve the quality of our article.

2) **Comment:** *In the optimization of the reaction conditions, did the authors try to further change the AgOAc loading? In the ESI, the table S2 shows the effect of solvents on this reaction. Can the authors explain the necessity of using solvent 1, which would be removed before adding substrate 2a and catalyst C9.*

Response: We sincerely thank reviewer 3 for the comments. We also tried to conduct the reaction using 5 mol% of AgOAc with 10 mol% of C9, while a declined yield was obtained (44% yield, 92:8 er).

Since the solvent may also affect the result of metal-catalyzed cycloisomerization of enynamide (such as the yield, reaction rate, and the ratio of (*E*)-**A** and *iso*-**A**), we screened different solvents for the first step and identified that CHCl₃ is also a good solvent for the cycloisomerization step.

3) **Comment:** *The substrate scope of the reaction is broad and sufficient. A range of MBH carbonates 2 were investigated by varying the substituents. However, only ethyl group substituted substrate at the C3 position of the pyrazolone moiety was examined, and what is the result of the reaction with an aryl group? In addition, the diastereoselectivities of all products should be provided in Table 2.*

Response: We sincerely thank reviewer 3 for the comment. Accordingly, we further investigated the substrate bearing a phenyl group at the C3 position of the pyrazolone moiety. However, the reaction did not give the desired product.

According to the suggestion, the diastereoselectivities of all products were provided in Table 2.

4) **Comment:** *The transformations of product 3a were also explored in this paper. How the authors explain the slight increase of enantioselectivity after formation of product 6 (from 95:5 er to 97:3 er). It seems to be weird because the stereogenic center, in general, is stable and hard to change.*

Response: We appreciate reviewer 3 for bringing up this point. We agree that the ee value of **6** seems to be outside the margin of error. Therefore, we repeated the reaction and obtained **6** in a similar yield with 94:6 er. The results and spectra were updated in the revised manuscript and SI.

At last, we would like to express our sincere gratitude to the reviewers for giving us the opportunity to revise and enhance our work. With your valuable feedback, we believe that we have adequately addressed all the concerns and improved the quality of our work significantly. We are confident that the revised manuscript meets the requirements for publication in *Communications Chemistry*.

REVIEWERS' COMMENTS:

Reviewer #2 (Remarks to the Author):

The manuscript seems to be appropriately revised based on reviewers comments, and this reviewer feels positive for this revised manuscript published in this journal.

Reviewer #3 (Remarks to the Author):

The revision has addressed the concerns from this reviewer. The manuscript is thus recommended for publication.